

# The immunostimulatory activity of polysaccharides from *Glycyrrhiza uralensis*

Adila Aipire[1], Mahepali Mahabati[1], Shanshan Cai[1], Xianxian Wei[1], Pengfei Yuan[1], Alimu Aimaier[1], Xinhui Wang[2] and Jinyao Li[1]

[1] Xinjiang Key Laboratory of Biological Resources and Genetic Engineering, College of Life Science and Technology, Xinjiang University, Urumqi, China
[2] College of Resource and Environment Sciences, Xinjiang University, Urumqi, China

## ABSTRACT

**Background**. The enhancement of immunity is very important for immunocompromised patients such as cancer patients with radiotherapy or chemotherapy. *Glycyrrhiza uralensis* has been used as food and medicine for a long history. *G. uralensis* polysaccharides (GUPS) were prepared and its immunostimulatory effects were investigated.
**Methods**. Human monocyte-derived dendritic cells (DCs) and murine bone marrow-derived DCs were treated with different concentrations of GUPS. The DCs maturation and cytokine production were analyzed by flow cytometry and ELISA, respectively. Inhibitors and Western blot were used to study the mechanism of GUPS. The immunostimulatory effects of GUPS were further evaluated by naïve mouse model and immunosuppressive mouse model induced by cyclophosphamide.
**Results**. GUPS significantly promoted the maturation and cytokine secretion of human monocyte-derived DCs and murine bone marrow-derived DCs through TLR4 and down-stream p38, JNK and NF-$\kappa$B signaling pathways. Interestingly, the migration of GUPS treated-DCs to lymph node was increased. In the mouse model, GUPS increased IL-12 production in sera but not for TNF-α. Moreover, GUPS ameliorated the side effect of cyclophosphamide and improved the immunity of immunosuppressive mice induced by cyclophosphamide. These results suggested that GUPS might be used for cancer therapy to ameliorate the side effect of chemotherapy and enhance the immunity.

## INTRODUCTION

The immune system including organs, cells and molecules plays critical roles in preventing infections, maintaining homeostasis and monitoring abnormal cells. Dendritic cells (DCs) are professional antigen presenting cells (APCs) and bridge the innate and adaptive immune responses, which capture foreign antigens and tumor antigens, process these antigens into peptides and display these peptides to naïve T cells through major histocompatibility complex (MHC) to induce antigen-specific immune responses in lymphatic tissues. For the induction of cellular responses, other signals need to be provided by DCs, such as co-stimulatory molecules and cytokines, which enhance the activation of naïve T cells

Corresponding author
Jinyao Li, lijinyao—@163.com

and promote the differentiation of activated T cells into the T helper (Th) cell subsets (*Kalinski, 2009*). Mature DCs secreted IL-12 direct the induction of Th1 responses and cytotoxic T lymphocytes (CTL) (*Carreno et al., 2013*; *Macatonia et al., 1995*). Moreover, mature DCs highly express CCR7, a chemokine receptor, which promotes DCs migration to the draining lymph node (LN) (*Randolph, Angeli & Swartz, 2005*) Therefore, DCs are pharmacological target of immunomodulatory agents including herbal medicines due to its critical role in the immune system (*Chen et al., 2006*; *Li, Li & Zhang, 2015a*).

Plant-derived polysaccharides have been drawn much attention due to their immunomodulatory activities and safety (*Kikete et al., 2018*; *Li, Li & Zhang, 2015a*). Accumulating evidence, including our own, has demonstrated that polysaccharides could enhance the immunity through activation of different targets, such as DCs and macrophages (*Ferreira et al., 2015*). Activation of the immune system by polysaccharides is mediated by pattern recognition receptors including scavenger receptors and toll-like receptors (TLRs) (*Ferreira et al., 2015*; *Li, Li & Zhang, 2015a*). It has been reported that polysaccharides can promote the maturation of DCs through TLR2/4 to upregulate the expression of co-stimulatory molecules and cytokine production (*Li et al., 2017b*; *Li et al., 2012*; *Li, Xu & Chen, 2010*; *Zhu et al., 2013*). *Glycyrrhiza uralensis* has been used as food and medicine, and contains various bioactive compounds including polysaccharides, triterpenes and flavonoids, and has anti-inflammatory and antioxidant activities (*Chen, Li & Gu, 2017*; *Yang et al., 2017*). We previously reported that the crude polysaccharides of *G. uralensis* enhanced the maturation and function of DCs (*Aipire et al., 2017*). Here, the polysaccharides were purified from *G. uralensis* and its immunostimulatory effects and the mechanisms were investigated.

## MATERIAL AND METHODS

### The preparation of *G. uralensis* polysaccharides (GUPS)

GUPS was purified using the DEAE-cellulose chromatography from the crude GUPS. Briefly, *G. uralensis* minced root was extracted with petroleum ether twice for 1 h, followed by the extraction with 80% ethanol twice for 1 h, then extracted with boiling water three times for 2 h. The supernatant was collected and concentrated using a rotary vacuum evaporator at 40 °C and decolorated with acticarbon, then the concentrated solution was precipitated twice with 4 volumes of ethanol at 4 °C for 24 h to obtain the crude GUPS. The crude GUPS dissolved in water was purified through DEAE-52 cellulose column and eluted with deionized water, 0.1, 0.2, 0.5 and 1 M NaCl solutions at 1.0 ml/min rate. The fractions eluted with 0.1 M NaCl were collected, lyophilized and named as GUPS. The polysaccharide content of GUPS is 93% and the molecular weight of GUPS is 29.1 kDa.

### Animals and ethics statement

BALB/c and C57BL/6 mice (6–8 weeks) were obtained from the animal center of Xinjiang Medical University (Urumqi, China) and housed in a temperature-controlled and light-cycled animal facility of Xinjiang University.

All animal experiments were approved by the Committee on the Ethics of Animal Experiments of Xinjiang Key Laboratory of Biological Resources and Genetic Engineering

 

(BRGE-AE001) and performed under the guidelines of the Animal Care and Use Committee of College of Life Science and Technology, Xinjiang University.

## DC induction and treatment

Bone marrow-derived DCs (BM-DCs) were induced by granulocyte macrophage-colony stimulating factor (GM-CSF, PeproTech) according to previously described (*Li et al., 2015b*). In brief, the femurs and tibias were isolated from C57BL/6 mice and bone marrow cells were flushed out with RPMI-1640. After centrifugation at 1,200 rpm for 7 min, the cells were collected and resuspended in RPMI-1640 supplemented with 10% heat-inactivated fetal bovine serum, 100 units/ml penicillin-streptomycin, 2 mM L-glutamine, 50 $\mu$M $\beta$-mercaptoethanol and 20 ng/ml GM-CSF at $1 \times 10^6$ cells/ml, and incubated in a 37 °C incubator with a 5% $CO_2$ atmosphere. On day 3, non-adherent cells were gently removed. Culture medium was changed every second day. On day 7, immature DCs were collected and treated with different concentrations (1, 5, 10, 20 and 50 $\mu$g/ml) of GUPS, or 20 ng/ml of LPS (Sigma-Aldrich) for 12 h. For the analysis of endocytosis, GUPS (20 $\mu$g/ml) and LPS treated DCs were inoculated with FITC-Dextran (Sigma-Aldrich) for 1 h and detected by flow cytometry. For analyzing the role of TLR2/4 and the downstream MAPK and NF-$\kappa$B signaling pathways, the inhibitors of 1 $\mu$M TAK-242 (MedchemExpress), 10 $\mu$M SB203658 (SB), 25 $\mu$M SP600125 (SP) and 10 $\mu$M U0126 (Cell Signaling Technology), and 10 $\mu$g/ml tosyl phenylalanyl chloromethyl ketone (TPCK, Beijing SolaiBao Technology, China), and 100 ng/ml TLR2 monoclonal antibody (mAb, InvivoGen) were used to pretreat DCs for 1 h, and then DCs were treated with 20 $\mu$g/ml of GUPS or LPS for 12 h.

For the induction of human monocyte-derived DCs (Mo-DCs), 30 ml of peripheral blood was collected from volunteers. The lymphocytes were separated by Human Lymphocyte Separation Medium (TianJinHaoYang Biological Manufacture, China), inoculated in culture-flask and incubated at 37 °C for 2 h. The non-adhered cells were removed and the adhered cells were cultured with GM-CSF (1,000 U/ml) and IL-4 (500 U/ml, PeproTech). On day 6, cells were collected and treated with 20 $\mu$g/ml of GUPS or 100 U/ml of TNF-$\alpha$ (PeproTech) for 18 h.

The phenotype of cells was analyzed by flow cytometry and cytokine production (IL-1$\beta$, IL-10, IL-12 and TNF-$\alpha$) in the supernatant was detected by ELISA using ELISA kit according to the manufacturer's instruction (Elabscience, China). Absorbance at 450 nm was measured using an ELISA plate reader (Bio-Rad, USA).

## DC migration in vivo

Untreated DCs or DCs treated with GUPS (20 $\mu$g/ml) were stained with carboxyfluorescein succinimidyl ester (CFSE) and named as DC+CFSE or GUPS-DC+CFSE, respectively, then these DCs ($1 \times 10^6$ in 50 $\mu$l PBS) were subcutaneously injected to the right flank. GUPS-DC without CFSE staining were used as negative control. After 24 h, both right and left inguinal lymph nodes (LNs) were isolated and lymphocytes were stained with APC-CD11c.

## Western blot

The antibodies including ERK, JNK, p38, NF-$\kappa$Bp65, IKK $\alpha$/$\beta$, I$\kappa$B $\alpha$, actin, histone and their phosphorylated antibodies were purchased from Cell Signaling Technology (USA).

Western blot was performed as described in our previous study (*Aipire et al., 2017*). Briefly, DCs were treated with GUPS (20 μg/ml) for 0, 10, 30, 60 and 240 min and proteins were extracted using Nuclear and Cytoplasmic Protein Extraction Kit (Beijing ComWin Biotech Co., Ltd). For inhibitor experiment, DCs were pretreated with 1 μM TAK-242 and then treated with GUPS (20 μg/ml) for 10 min. After separation with 12% SDS-PAGE, The target proteins were detected using the ECL assay kit (Beyotime Biotechnology Co., Ltd, China).

### In vivo mouse cytokine production
BALB/c mice were injected with 100 ng LPS or 100 μg GUPS in 100 μl PBS intraperitoneally and blood was collected from retro-orbital of mice after 3 h and 6 h. PBS (100 μl) was used as control. The levels of IL-12 and TNF-α in sera were quantified by ELISA. On day 3, mice were sacrificed, and spleens, thymuses, hearts and lungs were collected to take the photo.

### Immunosuppressive mouse model and drug administration
To induce immunosuppressive mouse model, the BALB/c mice were administered with 80 mg/kg cyclophosphamide (CTX, Jiangsu Shengdi Pharmaceutical Co., Ltd, China) intraperitoneally every day for 3 days. On day 4, mice were intraperitoneally treated with 20 mg/kg λ-carrageenan (λ-CGN, Sigma-Aldrich) or 20 mg/kg GUPS every day for 7 days and named as λ-CGN and GUPS groups, respectively. Mice treated with saline were named as model group. Naïve mice treated with saline were used as control group. Each group included 5 animals. Body weight of mice was monitored every two days. On day 11, mice were sacrificed and liver, kidney, spleen, thymus, heart and lung were taken to calculate organ indexes. Organ index (mg/g) = organ weight (mg)/body weight (g). Splenocytes were used to analyze the proportion and activation of immune cells.

### Flow cytometry
Fluorescein isothiocyanate (FITC)-, phycoerythrin (PE)-, allophycocyanin (APC)-labeled mAbs including anti-CD11c, anti-MHC-I/II, anti-CD8, anti-CD4, anti-CD44, anti-CD11b, were purchased from BD Biosciences (USA), and others including anti-CD40, anti-CD80, anti-CD86, and anti-CCR7 were bought from Elabscience (China). Cell surface staining was performed according to our previous description (*Li et al., 2016*). All samples were collected on FACSCalibur (BD Biosciences) and analyzed by the FlowJo platform (Tree Star, Inc., Ashland, OR).

### Statistical analysis
Data were reported as mean ± standard error of the mean (SEM). One-way analysis of variance (ANOVA), paired or unpaired $t$-test were used to analyze the statistical significance by Prism5 GraphPad software (GraphPad Software, La Jolla, CA). A value of $p < 0.05$ was considered to be statistically significant.

## RESULTS

### GUPS promotes the maturation of murine and human DC
To explore the effect of GUPS on DCs maturation, murine BM-DCs were prepared and treated with different concentrations of GUPS for 12 h. Compared to untreated BM-DCs,

GUPS significantly increased the expression of surface markers including CD40, CD80, CD86, MHC-I/-II and CCR7 (Figs. 1A–1F), and the secretion of cytokines including IL-1β, IL-12p40 and TNF-α (Figs. 1G–1I), in a dose-dependent manner. The effect of GUPS at high doses on BM-DCs maturation and cytokine production is even stronger than that of LPS. The effect of GUPS on human Mo-DCs was further detected. After treatment, GUPS significantly upregulated the expression of CD86 and HLA-DR compared with untreated Mo-DCs (Figs. 1J–1M). The effect of GUPS is similar with TNF-α. The supernatant was collected to detect the levels of IL-12 and TNF-α by ELISA. Compared to Untreated, both GUPS and TNF-α treatment significantly increased the level of IL-12. Moreover, the IL-12 level of GUPS treatment is higher than that of TNF-α treatment (Fig. 1N). GUPS treatment also increased the level of TNF-α (Fig. 1O). The above results indicated that GUPS promoted not only BM-DCs maturation but also Mo-DCs maturation.

With the maturation of DCs, the ability of antigen phagocytosis is decreased. Consistently, GUPS treatment significantly decreased the mean fluorescence intensity of FITC-dextran in BM-DCs compared to control BM-DCs (Fig. 2), suggesting that GUPS promoted BM-DCs maturation.

## GUPS enhances the migration of DC *in vivo*

CCR7 expression is essential for the migration of DCs to LN through interaction with chemokines CCL19 and CCL21, where DCs present antigen to naïve T cells and initiate cellular responses (*Worbs, Hammerschmidt & Förster, 2017*). The above data showed that GUPS significantly increased the expression of CCR7 on the surface of DCs. Therefore, the migration of GUPS-DCs was detected *in vivo*. GUPS-DCs and untreated DCs were stained with CFSE and injected into the right flank of mice. After 24 h, both sides of inguinal LNs were isolated to analyze the frequencies of CFSE$^+$ DCs. We observed that the frequencies of CFSE$^+$GUPS-DC were significantly increased compared with CFSE$^+$ untreated DC in both sides of LNs (Fig. 3), suggesting that GUPS treatment promoted the migration of DCs not only to the draining LN but also to the remote LN.

## TLR4 is involved in GUPS recognition

To investigate whether GUPS promotes DCs maturation via TLR4 and/or TLR2 signaling pathways, TLR4 inhibitor (TAK-242) and TLR2 mAb were used to pretreat DCs for 1 h before GUPS treatment. We observed that TAK-242 pretreatment significantly inhibited the expression of CD40 and CD86 (Figs. 4A–4J), and the production of IL-12 and TNF-α (Figs. 4K–4N) induced by GUPS, which is similar with LPS. However, TLR2 antibody pretreatment did not significantly suppress the expression of these molecules induced by GUPS and LPS. These results suggested that TLR4 might be the receptor on DCs for recognition of GUPS.

## GUPS activates NF-$\kappa$B and MAPK signaling pathways

The down-stream signaling pathway of TLR4 includes MAPK and NF-$\kappa$B molecules. Therefore, the activation status of molecules in MAPK and NF-$\kappa$B signaling pathways in DCs was detected by Western blot after treatment with GUPS. The phosphorylation of ERK, JNK, p38, IKK α/β, I$\kappa$B and NF-$\kappa$Bp65 was enhanced by GUPS treatment from 10

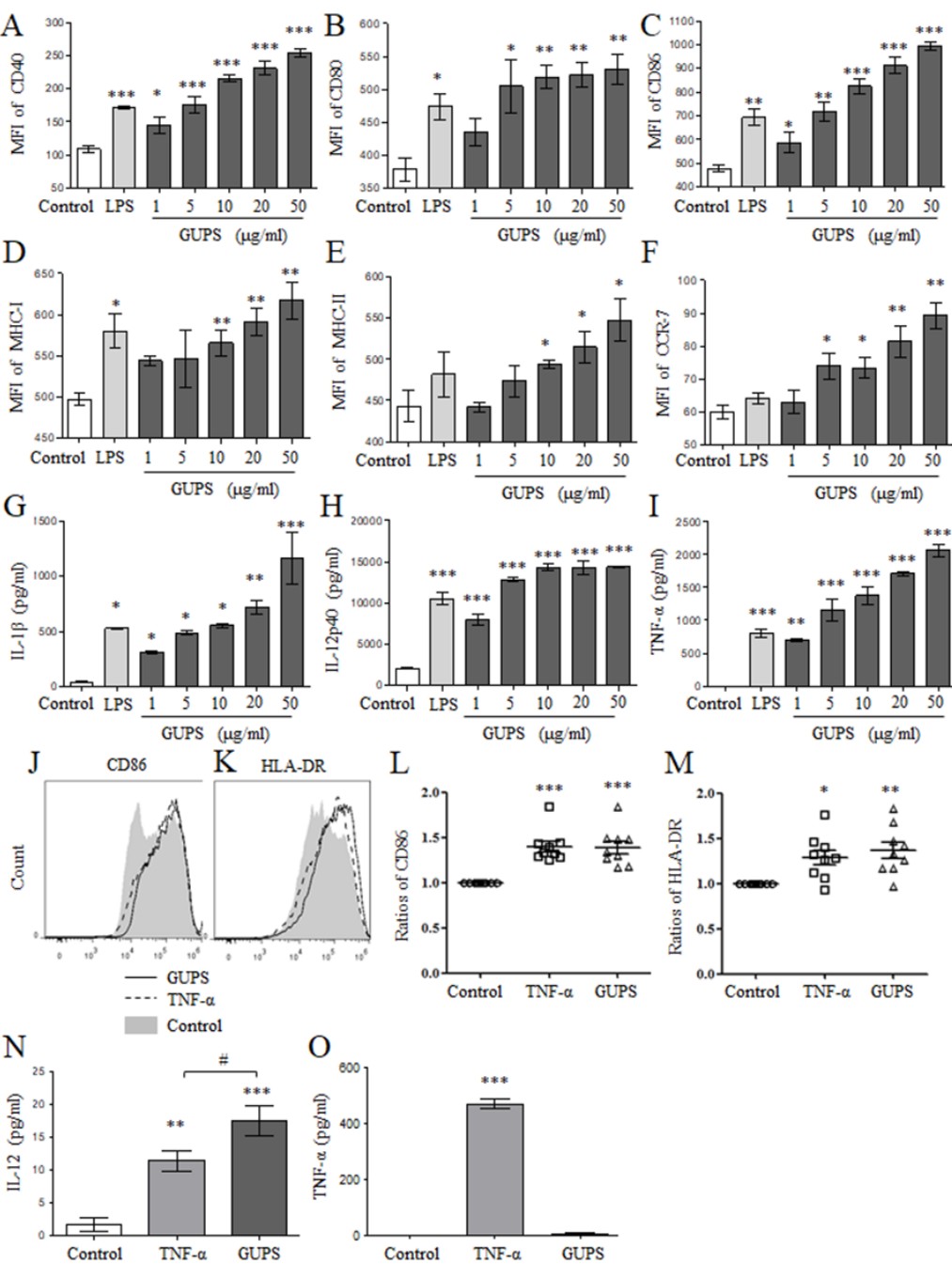

**Figure 1 The phenotype and cytokine secretion of DCs.** (A–F) BM-DCs were treated with different con-
centrations (1, 5, 10, 20 and 50 µg/ml) of GUPS for 12 h. The expression of surface molecules on DCs was
detected by flow cytometry. The mean fluorescence intensity (MFI) (mean ± SEM) of these molecules is
shown. (G–I) The cytokine production in supernatant of BM-DCs was detected by ELISA. Data are from 3
independent experiments. (J–L) Mo-DCs were treated with 20 µg/ml of GUPS for 18 h. The expression of
CD86 and HLA-DR on DCs was detected by flow cytometry. The MFI (mean ± SEM) of CD86 and HLA-
DR is shown. (M–N) The cytokine production in supernatant of Mo-DCs was detected by ELISA. Data
were analyzed by ANOVA. * $p < 0.05$; ** $p < 0.01$; *** $p < 0.001$ compared to untreated DC. # $p < 0.05$
compared to TNF- α treated DC.

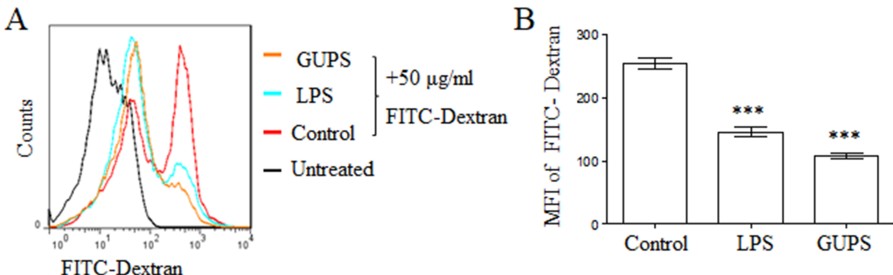

**Figure 2** **GUPS suppressed the phagocytosis of DCs.** DCs were treated with GUPS or LPS for 12 h, then co-cultured with FITC-Dextran for 1 h. (A) The fluorescence of FITC in DCs was analyzed by flow cytometry. (B) MFI of FITC was shown. Data were analyzed by ANOVA. *** $p < 0.001$ compared to Control.

min to 4 h (Fig. 5A). Although the phosphorylation of JNK was increased from 10 min to 60 min, the protein level of JNK was decreased from 30 min. The phosphorylation of NF-$\kappa$Bp65 arrived maximum at 4 h. Consistently, the level of NF-$\kappa$B in nuclei also arrived maximum at 4 h. TAK-242 was further used to examine the role of TLR4 in the activation of NF-$\kappa$B and MAPK signaling pathways. As shown in Fig. 5B, TAK-242 pretreatment partially inhibited the phosphorylation of ERK, JNK, p38, IKK $\alpha$/ $\beta$, I $\kappa$B and NF- $\kappa$Bp65, and decreased the level of NF-$\kappa$B in nuclei, suggesting that GUPS activated NF-$\kappa$B and MAPK signaling pathways through TLR4.

To demonstrate the role of NF-$\kappa$B and MAPK signaling pathways in DCs maturation induced by GUPS, the inhibitors SB, SP, U0126 and TPCK for p38, JNK, ERK and NF-$\kappa$B, respectively, were used to pretreat DCs to detect the maturation and cytokine production after GUPS treatment. As shown in Fig. 6, pretreatment with SB, SP and TPCK significantly suppressed the expression of CD40 and IL-12p40 induced by GUPS. Only TPCK pretreatment significantly decreased the expression of CD86 induced by GUPS. Interestingly, U0126 pretreatment did not affect the expression of CD40, CD86 and IL-12p40 but significantly decreased the production of IL-10 induced by GUPS, suggesting that ERK might be the potential target for regulation of DCs function. SB pretreatment also significantly decreased the production of IL-10 induced by GUPS. These results suggested that GUPS promoted DCs maturation and cytokine production through TLR4 and its downstream p38, JNK and NF-$\kappa$B signaling pathways.

## GUPS induces IL-12 production *in vivo*

We further detected the effect of GUPS on cytokine production in naïve mice. After intraperitoneal injection of GUPS or LPS, the cytokine profile in sera of mice was analyzed by ELISA. The level of IL-12 was significantly increased by GUPS, which is similar to LPS (Fig. 7A). Interestingly, GUPS did not induce inflammation in mice, whereas LPS induced inflammatory cytokine TNF-$\alpha$ production (Fig. 7B). On day 3, organs of mice were isolated and observed. LPS also showed the side effect on organs of mice, such as heart and thymus, which was not observed in GUPS-treated mice (Figs. 7C–7F). The results suggested that GUPS induced Th1 skewing cytokine profile *in vivo* without inflammation.

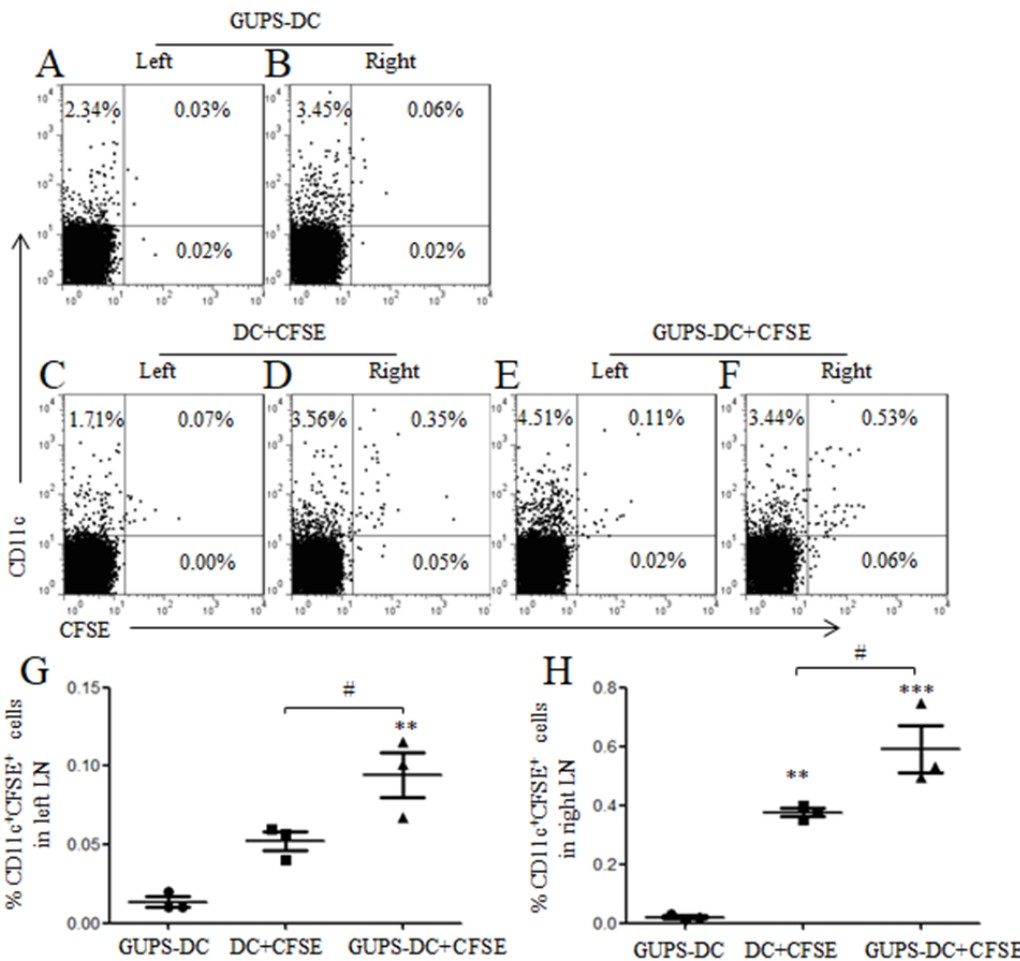

**Figure 3 DC migration *in vivo*.** DCs were treated with or without 20 $\mu$g/ml of GUPS and stained with CFSE. Cells were subcutaneously injected to the right flank. GUPS-DCs without CFSE staining were used as control. After 24 h, lymphocytes in both right and left inguinal LN were isolated. (A–F) CFSE $^+$ DCs were analyzed by flow cytometry. (G–H) Percentages of CD11c $^+$CFSE $^+$ cells in left and right LN were shown, respectively. Data are from 2 independent experiments and the representative data are shown and analyzed by ANOVA. ** $p < 0.01$; *** $p < 0.001$ compared to GUPS-DCs.

## GUPS enhances the immunity of immunosuppressive mice

CTX is widely used for tumor chemotherapy but it is adverse to immune system to cause immunosuppression. To test the immunostimulatory activity of GUPS, the immunosuppressive mouse model was induced by CTX. λ-CGN was used as positive control due to its immunomodulatory effect (*Bhattacharyya et al., 2010*; *Cheng, Chen & Chen, 2008*; *Li et al., 2017a*). After CTX injection, the body weight of mice was lost. Compared with model group, the body weight of mice was recovered by GUPS treatment, which is similar with λ-CGN treatment. On day 11, the body weight of mice in GUPS and λ-CGN groups was similar to the control group (Fig. 8A). On the same day, mice were sacrificed and organs were taken to calculate organ indexes. Compared to control group, the spleen and thymus indexes in model group were significantly reduced but the heart, kidney,

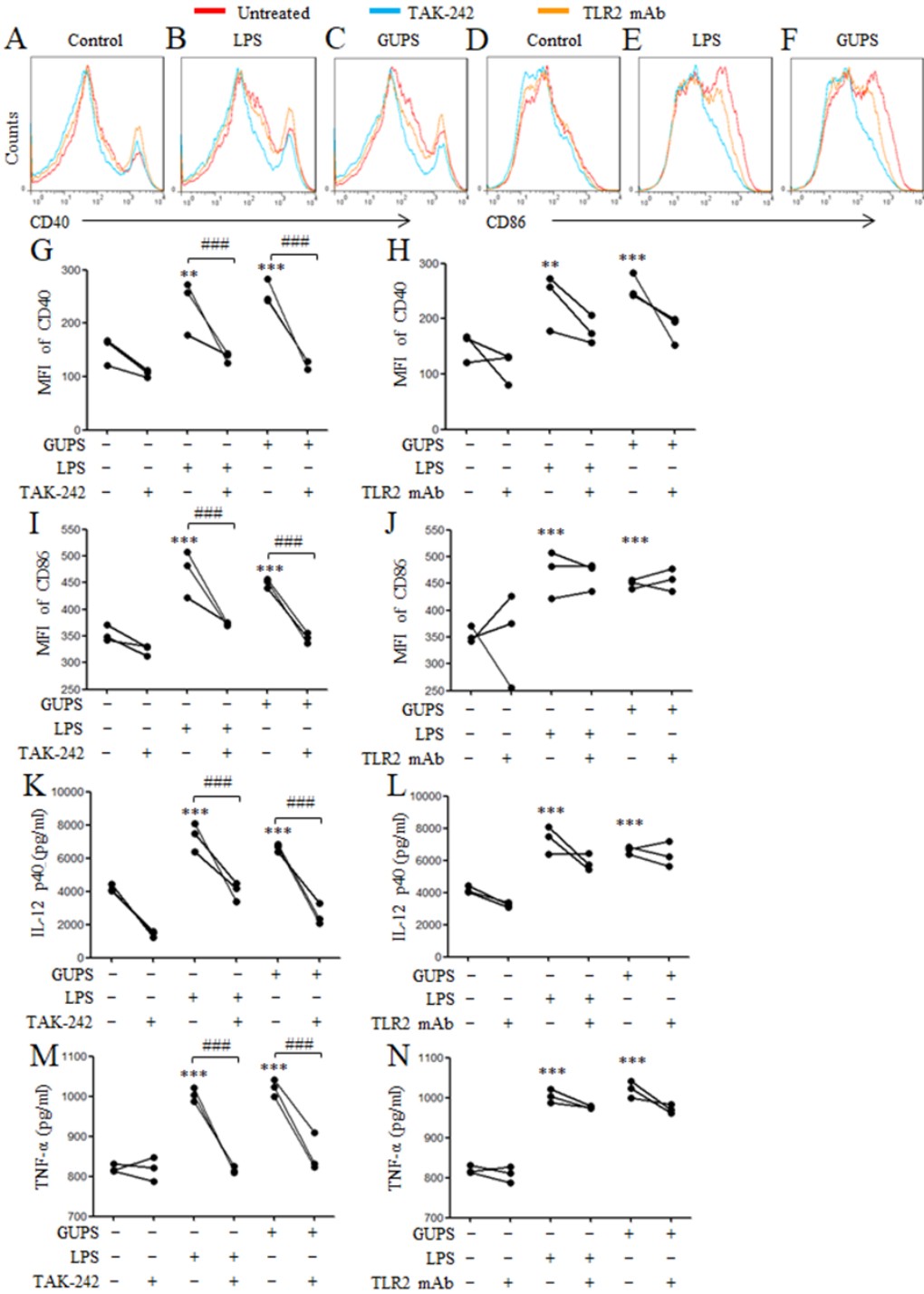

**Figure 4** **The effect of TLR4 and TLR2 blockade on DC maturation and cytokine production.** DCs were pretreated with or without 1 μM TAK-242 or 100 ng/ml TLR2 mAb for 1 h, and then treated with GUPS (20 μg/ml) or LPS for 12 h. (A–F) The expression of CD40 and CD86 were detected by flow cytometry. (G–J) MFI of CD40 and CD86 was shown. (K–N) The levels of IL-12 and TNF-α in supernatant were tested by ELISA. Data are from 3 independent experiments. ** $p < 0.01$; *** $p < 0.001$ compared to untreated DCs (ANOVA). ### $p < 0.001$ compared to TAK-242 treated DCs (paired $t$-test).

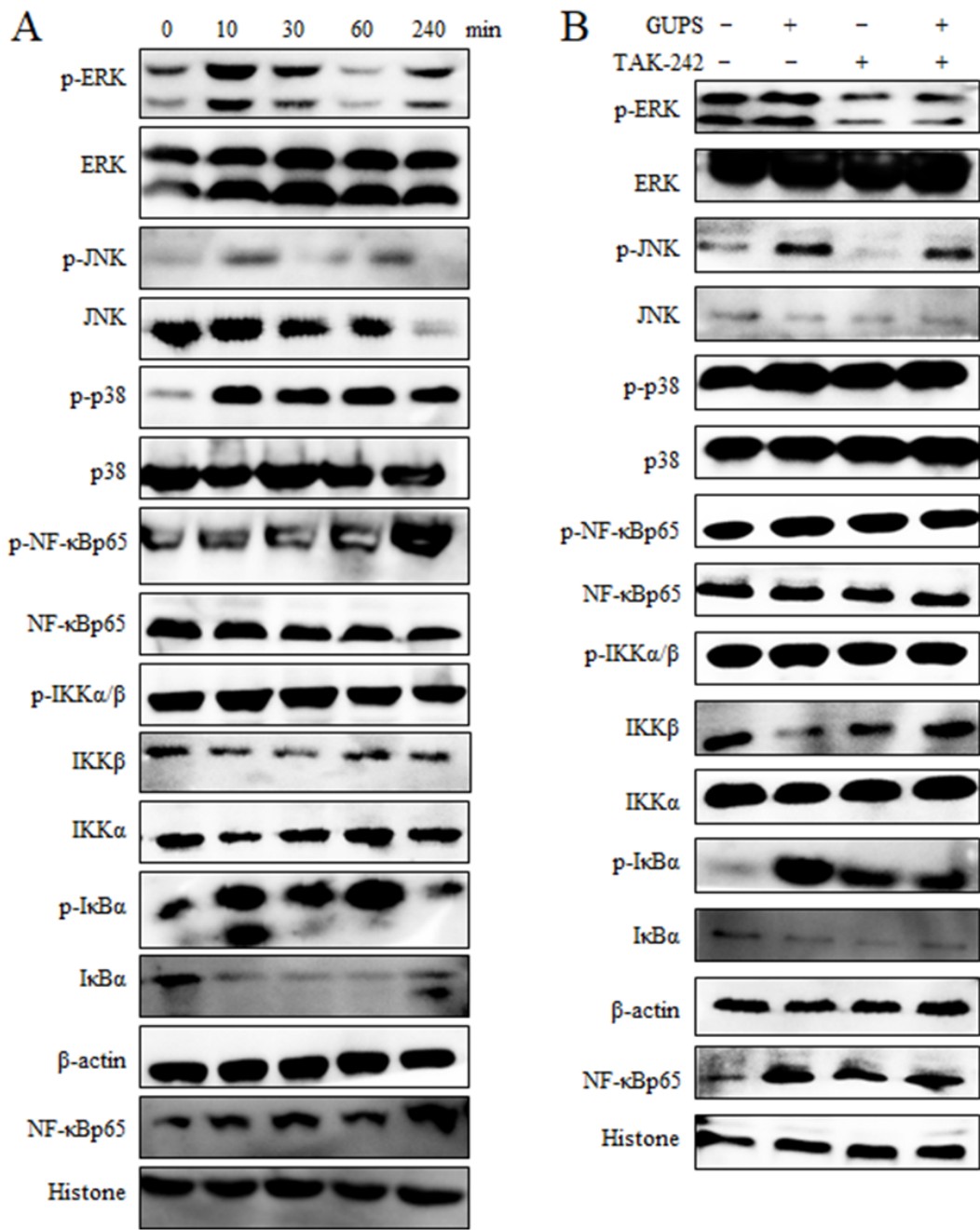

**Figure 5 The activation of MAPK and NF-κB signaling pathways.** (A) Nuclear and cytoplasmic proteins were isolated at the indicated time points from DCs treated with 20 μg/ml of GUPS. The levels of protein and their phosphorylation in cytoplasm or nuclei were detected by Western blot. (B) DCs were pretreated with TAK-242 for 1 h, then treated with 20 μg/ml of GUPS for 10 min. Nuclear and cytoplasmic proteins were prepared and detected by Western blot.

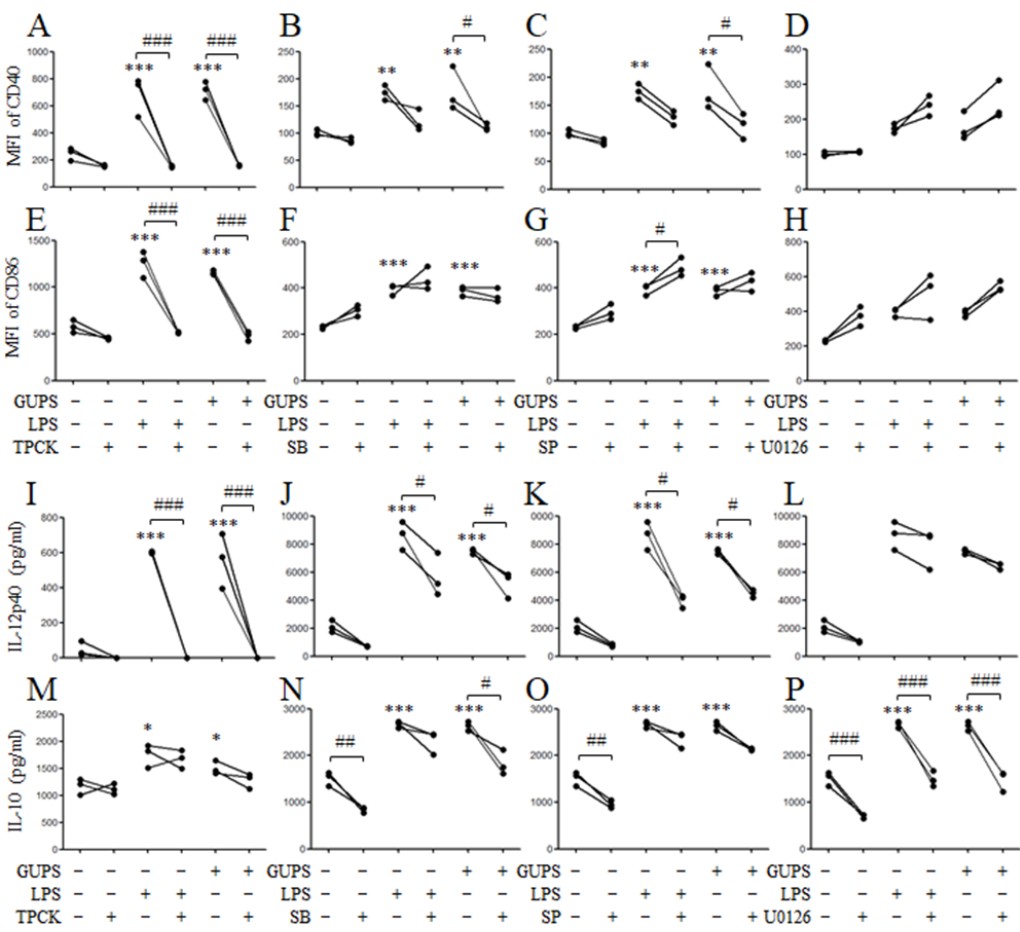

**Figure 6  The effect of MAPK and NF-κB blockade on DC maturation and cytokine production.** DCs were pretreated with or without 10 μg/ml TPCK, 10 μM SB, 25 μM SP or 10 μM U0126 for 1 h, and then treated with GUPS (20 μg/ml) or LPS for 12 h. (A–H) The expression of CD40 and CD86 were detected by flow cytometry. MFI of CD40 and CD86 was shown. (I–P) The concentrations of IL-12p40 and IL-10 in supernatant were measured by ELISA. * $p < 0.05$; ** $p < 0.01$; *** $p < 0.001$ compared to untreated DCs (ANOVA). # $p < 0.05$; ## $p < 0.01$; ### $p < 0.001$ compared to inhibitor treated DCs (paired $t$-test).

liver and lung indexes were partly increased (Table 1). Both GUPS and λ-CGN treatment recovered the spleen and thymus indexes compared with model group. GUPS treatment ameliorated the heart, kidney, liver and lung indexes but λ-CGN treatment aggravated these organ indexes compared with model group (Table 1). The results indicated that GUPS ameliorated the side effect of CTX.

The proportions of immune cells in spleens were further analyzed by flow cytometry. Compared to model group, GUPS and λ-CGN treatment significantly increased the proportions of CD3[+], CD11c[+] and CD11b[+] cells (Figs. 8B, 8E and 8F). The proportion of CD19[+] cells was also significantly increased by GUPS treatment (Fig. 8C). CD3[+] T cells are mainly composed of CD4[+] and CD8[+] T cells. We found that the proportion and activation (CD44[+]) of CD8[+] T cells were significantly improved by GUPS and λ-CGN

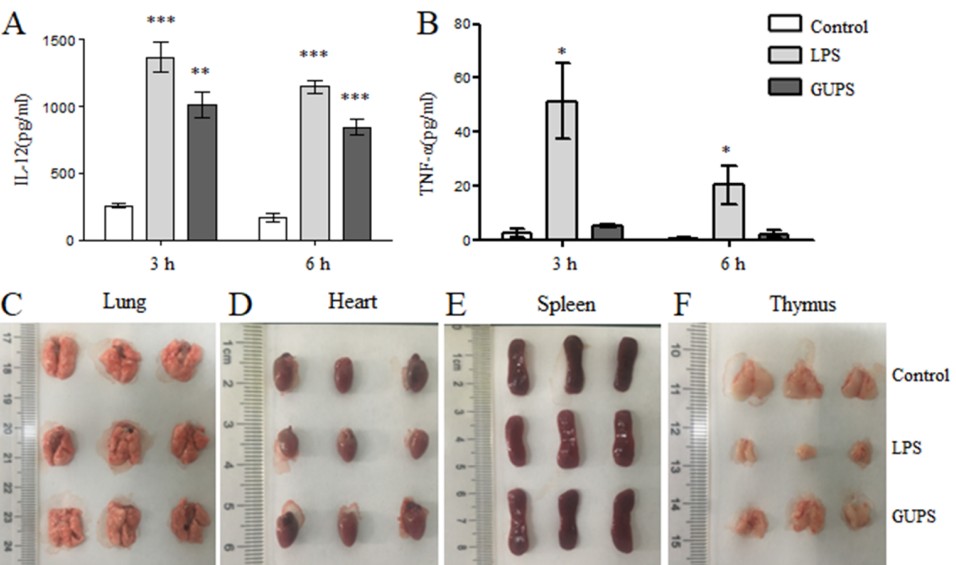

**Figure 7** **The effect of GUPS on naïve mice. Naïve BALB/c mice were injected with GUPS or LPS intraperitoneally.** (A–B) The production of IL-12 and TNF-α in sera after 3 and 6 h. (C–F) The organs of mice after 3 days of injection. * $p < 0.05$; ** $p < 0.01$; *** $p < 0.001$ compared to control group (ANOVA).

treatment (Fig. 8G 8I). The results indicated that GUPS not only increased the proportions of immune cells but also promoted the activation of CD8[+] T cells in immunosuppressive mice.

## DISCUSSION

Various plant-derived polysaccharides have been isolated and shown immunostimulatory activity (*Kikete et al., 2018*). Here, we found that GUPS promoted the maturation of DCs through TLR4 and its downstream p38, JNK and NF-κB signaling pathways, increased IL-12 production *in vivo* and enhanced the immunity of immunosuppressive mice.

The interaction of co-stimulatory molecules among DCs and T cells is necessary for the full activation of naïve T cells (*Kalinski, 2009*). Lots of studies including ours have shown that polysaccharides can promote DC maturation characterized by the upregulated expression of co-stimulatory molecules (*Du et al., 2012*; *Kim et al., 2011b*); Li et al. 2017; (*Tian et al., 2014*). Similarly, GUPS promoted the maturation and cytokine production of mouse BM-DCs and human Mo-DCs, especially IL-12 production that facilitates the differentiation of Th1 response (*Carreno et al., 2013*; *Macatonia et al., 1995*). Th1 response and CTL play important roles in the defense of infection and tumor, suggesting that GUPS might be used as adjuvant for the development of vaccine or immunostimulatory agent to treat immunosuppressive or cancer patients. Further, GUPS increased IL-12 production in mice without induction of TNF- α and side effect on organs. This was different with LPS, which could induce inflammation (*He et al., 2017*; *Kim et al., 2018*). It has been reported that LPS can induce thymocyte death and thymic atrophy (*Huang et al., 2016*). Similarly, we also observed the thymic atrophy induced by LPS, which was not found in GUPS

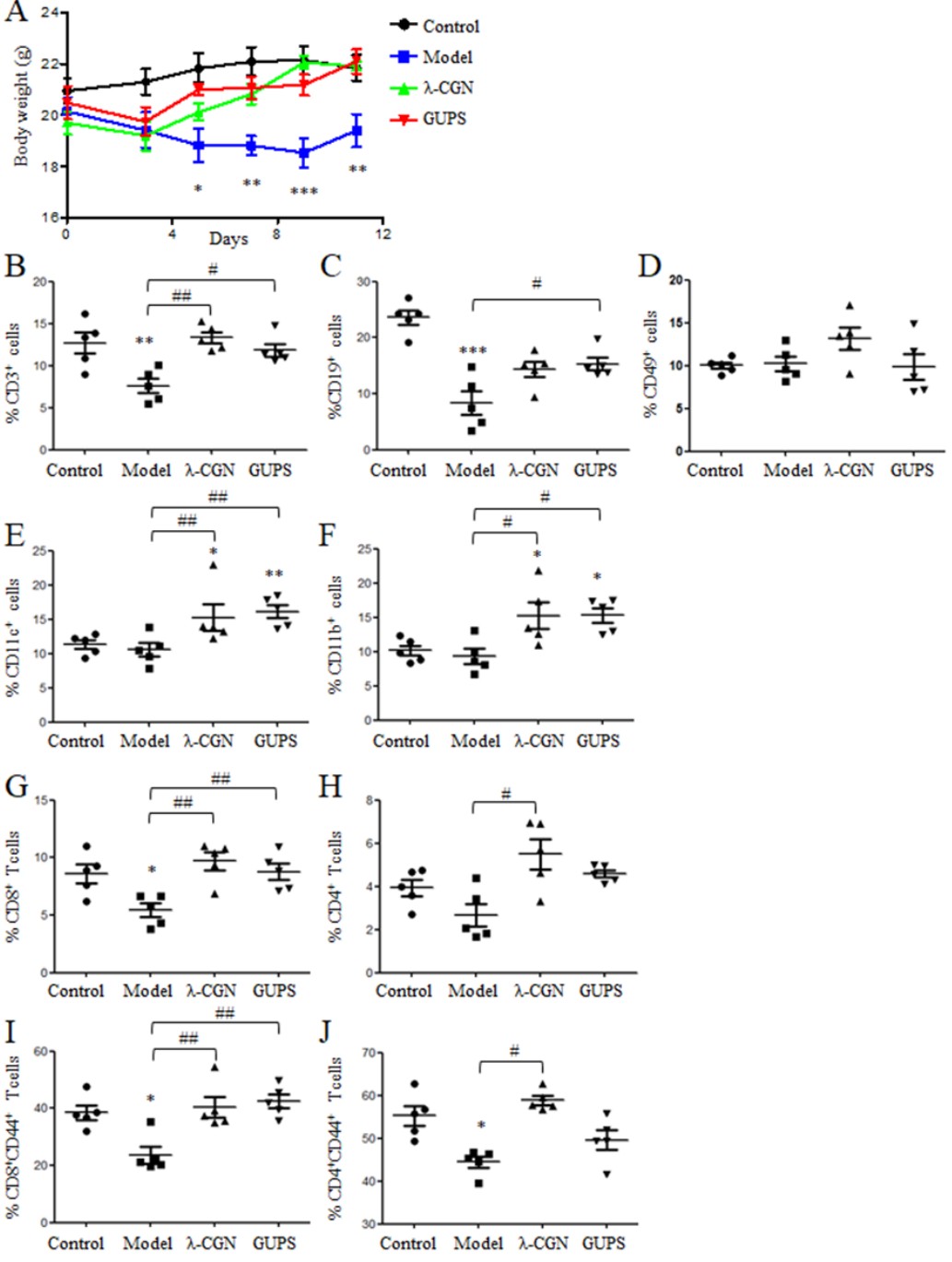

**Figure 8 The effect of GUPS on immunosuppressive mouse model.** Immunosuppressive mouse model was induced by CTX and treated with GUPS or λ-CGN. (A) The body weight of mice. (B–F) The proportions of immune cells in spleens of mice. On day 11, spleens were isolated to analyze the proportions of immune cells by flow cytometry. (G–J) The proportions and activation of CD4$^+$ and CD8$^+$ T cells in spleens of mice. ** $p < 0.05$; ** $p < 0.01$; *** $p < 0.001$ compared to control group (ANOVA). # $p < 0.05$; # # $p < 0.01$ compared to model group (ANOVA).

**Table 1 Organ indexes of mice.**

| Organ index (mg/g) | Control | Model | λ-CGN | GUPS |
|---|---|---|---|---|
| Spleen index | $4.77 \pm 0.85$ | $3.62 \pm 1.24$ | $11.60 \pm 4.00$**/### | $9.54 \pm 2.02$*/## |
| Thymus index | $3.31 \pm 0.81$ | $1.13 \pm 0.56$*** | $2.29 \pm 0.86$# | $2.34 \pm 1.09$## |
| Heart index | $4.92 \pm 1.09$ | $5.81 \pm 0.95$ | $6.26 \pm 0.41$ | $5.22 \pm 0.62$ |
| Kidney index | $13.36 \pm 1.42$ | $13.95 \pm 1.82$ | $16.48 \pm 0.93$* | $13.21 \pm 2.02$ |
| Liver index | $51.22 \pm 3.04$ | $55.74 \pm 6.19$ | $67.38 \pm 4.11$**/# | $53.85 \pm 7.23$ |
| Lung index | $7.89 \pm 0.71$ | $9.02 \pm 2.62$ | $9.76 \pm 1.07$ | $8.76 \pm 0.46$ |

Notes.
*$p < 0.05$.
**$p < 0.01$.
***$p < 0.001$ compared to control group (ANOVA).
#$p < 0.05$.
##$p < 0.01$.
###$p < 0.001$ compared to model group (ANOVA).

group. The results indicated that GUPS had the potential to be developed as an adjuvant or immunostimulatory agent.

The immunostimulatory effect of GUPS was evaluated in immunosuppressive mouse model induced by CTX. Several studies have been demonstrated that natural polysaccharides from various sources can ameliorate immunosuppression induced by CTX (*Chen et al., 2012*; *Wang et al., 2011a*; *Wang et al., 2012*). Similarly, GUPS treatment recovered the proportions of immune cells in spleen of mice. Moreover, GUPS ameliorated the side effects of CTX on body weight and organs. This was different with λ-CGN, which aggravated organ indexes compared with model group. The results suggested that GUPS might be used as immunostimulatory agent for cancer therapy to ameliorate the side effects of chemotherapy and improve the efficacy.

It has been reported that polysaccharides can stimulate immune cells through different types of receptors including TLRs and scavenger receptors, which sequentially activate MAPK and NF-$\kappa$B signaling pathways (*Ferreira et al., 2015*; *Kikete et al., 2018*). Our previous studies showed that *Pleurotus ferulae* polysaccharides and *G. uralensis* water extract promoted DCs maturation and cytokine production through TLR4 signaling pathway (*Aipire et al., 2017*; *Li et al., 2017a*; *Li et al., 2017b*). Here, we observed that GUPS promoted DC maturation through TLR4 and activated its downstream signaling pathways such as MAPKs and NF-$\kappa$B. The inhibitors of MAPKs and NF-$\kappa$B were further used to investigate their roles in the expression of co-stimulatory molecules and cytokines. We found that NF-$\kappa$B inhibitor blocked the expression of CD40, CD86 and IL-12 induced by GUPS. p38 and JNK inhibitors partially inhibited the expression of CD40 and IL-12 but did not affect the expression of CD86 induced by GUPS. Interestingly, ERK inhibitor suppressed the production of IL-10 but did not change the expression of CD40, CD86 and IL-12 induced by GUPS. These results provided the details of the MAPK and NF-$\kappa$B signaling pathways that involved in the DCs maturation and cytokine production induced by the plant-derived polysaccharides, and the potential target, ERK, for optimizing DCs function. The ideal DCs for vaccine should secrete high level of IL-12 and low level of IL-10. Several studies have shown that the immune responses induced by DC-based vaccines can

be improved through downregulation of IL-10 production and upregulation of IL-12 production (*Kim et al., 2011a*; *Wang et al., 2011b*). Therefore, the combination of GUPS and ERK inhibitor might be used to prepare DC-based vaccine.

## CONCLUSION

GUPS promoted the maturation of DCs through TLR4 and its downstream p38, JNK and NF-$\kappa$B signaling pathways. GUPS promoted Th1 skewing cytokine profile *in vivo* and enhanced the immunity of immunosuppressive mice. GUPS might be used as an effective immunostimulatory agent for cancer therapy.

### Funding

This work was supported by the National Natural Science Foundation of China (U1803381 to Jinyao Li and 31760260 to Xinhui Wang), the 2017 Doctoral Start-up Fund of Xinjiang University and the 1000 Young Talents Program of China to Jinyao Li, and the Graduate Science and Technology Innovation Project of Xinjiang University (XJUBSCX-2016016) to Adila Aipire. The funders had no role in study design, data collection and analysis, decision to publish, or preparation of the manuscript.

### Grant Disclosures

The following grant information was disclosed by the authors:
National Natural Science Foundation of China: U1803381, 31760260.
2017 Doctoral Start-up Fund of Xinjiang University.
1000 Young Talents Program of China.
Graduate Science and Technology Innovation Project of Xinjiang University: XJUBSCX-2016016.

### Competing Interests

The authors declare there are no competing interests.

### Author Contributions

- Adila Aipire conceived and designed the experiments, performed the experiments, analyzed the data, prepared figures and/or tables, authored or reviewed drafts of the paper, and approved the final draft.
- Mahepali Mahabati, Shanshan Cai, Xianxian Wei, Pengfei Yuan and Alimu Aimaier performed the experiments, prepared figures and/or tables, and approved the final draft.
- Xinhui Wang analyzed the data, prepared figures and/or tables, and approved the final draft.
- Jinyao Li conceived and designed the experiments, analyzed the data, prepared figures and/or tables, authored or reviewed drafts of the paper, and approved the final draft.

## Animal Ethics

The following information was supplied relating to ethical approvals (i.e., approving body and any reference numbers):

All animal experiments were approved by the Committee on the Ethics of Animal Experiments of Xinjiang Key Laboratory of Biological Resources and Genetic Engineering (BRGE-AE001).

## Data Availability

The raw measurements are available in a Supplementary Files.

## Supplemental Information

Supplemental information for this article can be found online at http://dx.doi.org/10.7717/peerj.8294#supplemental-information.

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
