# Peer review of "The immunostimulatory activity of polysaccharides from Glycyrrhiza uralensis"

_PeerJ, doi:10.7717/peerj.8294_

## Round 0.1 · original submission · Major Revisions

Dear Dr. Li,

In light of the feedback of the authors, I ask you to thoroughly address the drawbacks listed within the comments of the reviewers. Specifically, language, experimental design, and consistency should be improved and discussed. Please see the reviewer's comments for further details.

Reviewer 1 ·

Basic reporting

The study from Aipire et al. investigates the immunostimulatory effect of Glycyrrhiza uralensis in vitro and in murine vivo studies and is definitely worthy of publication. The experiments carried out, fit into the logic of argumentation and are all in all well performed. Minor changes have to be done to enhance its quality.

Some general English language changes have to be made.

- The authors should think about changing DC into DCs if they are using the plural form of multiple cells to increase reading fluency.

Specific changes:
29: add an article to "mouse"
28: add an article to "immune" and a comma before "and" (same error is repeated several times)
49: "therefore DCs are pharmacological targets of immunostimulatory agents..."
108: article to "supernatant"
139: strange sentence --> rearrange or add an article to "photo"
200: add article
add several articles to nouns in the whole text! there are several errors!

Experimental design

Materials and methods:

- please mention how you isolated the cells where the BM-DCs were derived from with GM-CSF and also add information about the culture conditions.
- where did side-effects observe in mice injected with CTX?
- be accurate: you show +SEM and SD and not +/- SEM and SD as described under statistics.
- what LPS concentration was used for the in-vivo studies?
- please add the number of replicates and what is presented to the figure legends, also mention if data are representatives (fig. 4?)
- what t-test did you utilize, because 5 animals per group may not fulfill test requirements

Results:

- please explain the different purpose for the use of BM-DCs and Mo-DCs
- specify the difference between control and untreated
- you only used 20ng LPS but startet from 1 µg GUPS --> discuss the differences in the dose dependent effects
- fig. 3: % negtive is a rare cell analysis and maybe the MFI of CFSE would also be helpful. Alternative: show unstained T-cells and start the gating from these population so that no junk is included (as seen at 10^0?!)
- fig. 6: highlight key results and important changes!
- fig. 8 and table 1: you have to discuss the smaller thymusses in the LPS group. Why are they smaller than the control thymuses?
- fig. 9a: color-coding would be helpfull (rethink that for some other figures too)

Figure legends:

- put more attention to the figure legends (see above)

Validity of the findings

Discussion and conclusion:

- you mention signaling pathways but only investigated some factors. Explain how they interfere with each other or make an overview.

The experiments you carried out are not enough to conclude the therpy as "without side-effects" and as "safe and effective".
- please specify that you only have hints regarding safety levels
- specify the use for cancer treatment

To better clarify the use of the study the authors
should discuss the possible clinical application of Glycyrrhiza uralensis:
- possible side-effects or hyper-inflammation (septic shock, application routes?)
- possible other TLR agonists as LPS or imiquimod, that is currently used in the therapy of cancerogenious lesions.

Additional comments

For further investigations, it would be helpful to investigate the M1/M2- polarization of macrophages, as a very relevant setting for immunity and tumor defense.

Reviewer 2 ·

Basic reporting

There is sometimes a lack of information to understand the reasons for the following experiments.

Experimental design

In general, the initial observations of this paper are very interesting, however, not to be despised part of the data is interpreted incorrectly or the experimental planning is partly not well designed.
The other major drawbacks of the paper are the superficialities or inconsistencies.

Validity of the findings

The validity of the data and statements can be improved. Based on the data too much is speculated.

Additional comments

Please, work on the experimental design and care of your interpretation and presentation of the data.

Annotated reviews are not available for download in order to protect the identity of reviewers who chose to remain anonymous.

---

## Round 0.2 · Minor Revisions

The specific criticism was reasonably addressed.

However, the gating of the CFSE experiment is highly insufficient.

The entire experiment should be either removed if unsure on the gating or re-evaluated.

I see the following problems here:

- junk was included in the forward vs side scatter gate
- therefore, the CD8 cells probably contain junk, this can be curated by gating on the central red population, and then gate on CD8
- there is no positive population for CD4, why? this population should be proliferating, too
- the histogram shows the insufficiency of the data. first, why is there is second peak at the right of the large peak? is this duplicate cells? second, proliferating T-cells - assuming they proliferate once per day so three cycles within 72h (your incubation time) - should have ~12.5% of the original signal. This would be approximately at about 1x10E2 on your scale. However, within the first two logs left of your main peak there is no proliferating cells. What you instead "quantify" is the cells at the far left because they are junk that has not been gated out in the first step. Either way, the CFSE gate should not go until the far left but rather the first three decades left of the main peak because this then contains already ~6-8 proliferation cycles (and more is rarely observed). as i do not see such population in any of the histograms, I would recommend to give the percentages not including the very left axis even if that leads to lack of changes.

If you claim to do see differences in CFSE, attach the respective .fcs files to the next round of revision so I can double check.

---

## Round 0.3 · accepted · Accept

The remaining comments were sufficiently addressed.